# What Factors Affect Binocular Summation?

**DOI:** 10.3390/brainsci14121205

**Published:** 2024-11-28

**Authors:** Marzouk Yassin, Maria Lev, Uri Polat

**Affiliations:** School of Optometry and Vision Sciences, Bar-Ilan University, Ramat Gan 5290002, Israel; marzooq.yasin@gmail.com (M.Y.); maria.lev@biu.ac.il (M.L.)

**Keywords:** collinear facilitation, lateral masking, binocular vision, binocular summation, spatial interactions, inter-ocular suppression, binocular advantage, stereo glasses, awareness, presentation time

## Abstract

Binocular vision may serve as a good model for research on awareness. Binocular summation (BS) can be defined as the superiority of binocular over monocular visual performance. Early studies of BS found an improvement of a factor of about 1.4 (empirically), leading to models suggesting a quadratic summation of the two monocular inputs (√2). Neural interaction modulates a target’s visibility within the same eye or between eyes (facilitation or suppression). Recent results indicated that at a closely flanked stimulus, BS is characterized by instability; it relies on the specific order in which the stimulus condition is displayed. Otherwise, BS is stable. These results were revealed in experiments where the tested eye was open, whereas the other eye was occluded (mono-optic glasses, blocked presentation); thus, the participants were aware of the tested eye. Therefore, in this study, we repeated the same experiments but utilized stereoscopic glasses (intermixed at random presentation) to control the monocular and binocular vision, thus potentially eliminating awareness of the tested condition. The stimuli consisted of a central vertically oriented Gabor target and high-contrast Gabor flankers positioned in two configurations (orthogonal or collinear) with target–flanker separations of either two or three wavelengths (λ), presented at four different presentation times (40, 80, 120, and 200 ms). The results indicate that when utilizing stereoscopic glasses and mixing the testing conditions, the BS is normal, raising the possibility that awareness may be involved.

## 1. Introduction

In normal vision, binocular summation (BS) refers to the phenomenon in which visual performance is greater when both eyes are used together compared to when each eye is used separately [1,2,3]. According to research findings, the ability to detect contrast and luminance thresholds is typically 40–60% better via binocular compared to monocular viewing [4,5,6,7,8,9,10,11]. These empirical findings led to models suggesting a quadratic summation of the two monocular inputs (√2). Interestingly, recently, it has been reported that BS can be influenced by crowding, tagging [12], and context [13].

Several models [14,15,16,17,18,19,20,21,22] of BS have been elaborated for isolated stimuli [23,24,25,26,27,28,29,30,31] in the last few decades, whereas other models [32,33,34,35,36,37,38,39,40,41,42,43,44,45,46,47,48,49,50,51,52,53,54] of BS have been elaborated for stimuli with context [42,55,56,57,58,59,60,61,62,63,64,65,66,67,68]. Early studies of BS found an improvement of a factor of about 1.4, leading to models suggesting a quadratic summation of the two monocular inputs (√2) [3,17,18,69,70,71,72]. In contrast sensitivity (CS) [31], BS refers to the equation of CSbin = sqrt (CSright2 + CSleft2) [19,73]. A recent review showed that different amounts of BS exist in a range from √2 to 2 values [15] because the amount of BS is affected by the spatiotemporal parameters of the stimulus. In the last few decades, several models of BS have been proposed (see Refs. [14,15,16,17,18,74,75,76,77,78,79,80,81,82]). Studies suggested a gain control theory [16,80,81,83] based on studies by Cogan [18] and Wilson [21]. They suggested that each eye exerts gain control on the other eye’s signal in proportion to the contrast energy of its own input; moreover, each eye exerts gain control on the other eye’s gain control [16,17,20]. Other studies suggested models with gain enhancement [84,85], suggesting that the contrast detection facilitation at threshold levels induced by cross-orientation masks with a single free parameter for gain enhancement across all spatiotemporal conditions and eyes. Baker et al. [3] defined a measure of “stimulus speed” that they calculated as the ratio between stimulus temporal (the presentation time) and the spatial frequency. According to this ratio, a slow speed (including high spatial and low temporal frequencies) will lead to a higher BS [3].

Hubel and Wiesel [86,87] explored the classical receptive fields (CRFs) of simple cells in the V1 area of the visual cortex. They found that the cells tuned selectively for location, orientation, and spatial frequency, forming the fundamental units of analysis. In V1, neurons are sensitive to specific stimulus features such as location, orientation, and the spatial frequency of the object presented in their RF. It was found that RF processing is affected by feedforward stimuli and lateral interactions [88,89] (excitatory or inhibitory signals via the horizontal connections between the ocular dominance columns found in V1) as well as by feedback processing.

Retinal ganglion cells (RGCs) and lateral geniculate nucleus (LGN) receptive fields cannot be defined in terms of spatial frequency; the best way to define the size of the receptive field in physiology is in terms of the visual angle. In an early study of lateral interactions in humans [90], it was demonstrated that even when using two Gabor functions and doubling their spatial frequency, the size of the perceptive field in the visual angle is doubled, but the maximal facilitation is still found at collinear three wavelengths (λ). This relationship was confirmed in many later studies. A later physiological study [91] in a cat’s V1 confirmed this finding; different sizes of the receptive fields that were measured in visual angles show facilitation, in agreement with the psychophysical studies. Therefore, in this study, the spatial distances between the Gabor target and flankers are measured in wavelength (λ) units.

Lateral interactions are believed to play a role in contextual modulation, affecting both masking and crowding [90,92,93]. Lateral interactions [94,95] involve excitation (facilitation) or inhibition (suppression) by neighboring neurons. These neurons, which share neighboring classical receptive fields (the receptive field serves as the basic processing unit in human vision), will also be physically near to each other, due to the topography of the primary visual cortex (V1) via long-range horizontal connections [88,96,97] between similar orientation columns [89,98,99,100,101] within the V1 area [95,102,103].

The effect of lateral interactions can be revealed by visual masking experiments [104] by modulating the visual response and contrast [90,94,105,106]. In such experiments (the lateral masking paradigm, LM), the detection of a Gabor patch target can either be enhanced (facilitation) or suppressed (suppression) based on the distance from high-contrast collinear flankers [92,93,94,106,107]. Here, there is a combination of two types of spatial interactions, namely, suppression and/or facilitation, relying on whether the target–flanker separations are either at two or three wavelengths (λ). Facilitation for collinear configuration is maximal when the flankers are separated from the target by 3λ, and it decreases for longer distances [92,93,108,109]. However, the contrast detection thresholds of a collinear target–flanker separation of 2λ can be elevated (suppression), especially in naïve participants [13,110,111].

When measuring BS, one should consider the lateral interactions at the binocular level. In the LM experiments, there are two types of suppressive processing: inter-ocular (between eyes) and local (monocular within each eye).

Interactions from the vicinity of the receptive field, in the context of physiology and psychophysics, are usually related to the interactions within the receptive field [88,89,98,99,100] (the perceptive field in psychophysics) and considered as local. Therefore, the local interactions may be valid if the interactions were induced by another receptive field [13,92,93]. In this sense, lateral inhibition is largely considered as local. In contrast, there are long-range interactions between neurons that may be mainly excitatory [90,94,95,96,97,106]. Considering these notions, local inhibition (suppression) [102,103,104,105,106,107,108,109,110] may be found mainly at the monocular level, whereas suppression between eyes (inter-ocular) is considered at a longer range and, thus, not local (between ocular dominance rather than within ocular dominance).

Thus, at close distances (collinear 2λ), there are two types of suppression: one caused by lateral interactions and the other one caused by inter-ocular suppression. When both types of suppression act together, detection is suppressed more under binocular than under monocular presentation [13,110,111,112]. In contrast, greater distances (collinear 3λ) facilitate both monocular and binocular detection, due to the collinear facilitation at 3λ. A recent study [13] indicated that the monocular lateral interactions are processed before the binocular integration.

### 1.1. BS Utilizing Mono-Optic Glasses

In our previous study [110], we utilized mono-optic glasses to control the monocular and binocular presentations (vision) by covering one eye with mono-optic glasses; the participants were aware of the eye that received the input.

Interestingly, we discovered that the sequence of the stimulus presentation time influences the binocular interactions and BS. The main results indicated that there was no BS at close distances (collinear 2λ) because the suppression of target detection is greater under binocular than under monocular presentation. However, BS exists for greater distances at collinear configurations (3λ because there is facilitation) or for orthogonal configurations (2λ and 3λ because there is no effect of suppression or facilitation). Importantly, although initially lateral suppression exists at collinear 2λ, which reduces or eliminates BS, the order in which the stimulus conditions are tested has an impact on both binocular interactions and BS, causing reduced lateral suppression at collinear 2λ; hence, this restores BS. The dynamics in the collinear interactions relies on the sequence of the stimulus presentation time and on the testing order of the stimulus condition. The order of the stimulus presentation time influences the suppression at 2λ and, consequently, the BS. Namely, it relies on whether the stimulus was displayed from the shorter to the longer presentation time or vice versa or if the presentation times were mixed. These results led us to hypothesize that the participant’s awareness of the tested eye might affect the induced inter-ocular suppression from the open eye on the covered eye. Our starting point stems from the assumption that participants have no knowledge of the eye of origin. Therefore, when dichoptic glasses are used, this is the case. However, when one eye is occluded, there is a lack of information that is received from the occluded eye, and, sometimes, participants perceived rivalry between eyes (which is why we do not use a black occlude). This phenomenon may suggest that some sort of awareness may lead to this rivalry and may reduce the BS.

### 1.2. BS Utilizing Stereo Glasses

It has been suggested [3] that the best way to measure BS is by utilizing equipment such as stereoscopes, shutter goggles, or virtual reality designed for binocular presentation. An interesting question that was raised from the data is whether the observers are aware of the stimuli shifting between the eyes. Herein, we explored whether the testing order influences the stimulus condition such as “the mixed between eyes” condition. More specifically, we utilized dichoptic googles, in which the participants were unaware of the eye that perceives the stimuli. Thus, an open question that might be addressed is whether BS may be affected by additional factors, such as awareness or long-term persistence of collinear facilitation from 3λ, which can persist with mixed trials involving different eyes, thus decreasing the suppression at collinear 2λ [90,94,95,106,108,113].

In the current study, we aimed to compare the results using two methods: utilizing mono-optic glasses as in our previous study [110] vs. stereo glasses (the current study). This information provides valuable insights to better understand the mechanisms underlying BS. We hypothesized that a relatively rapid alternation between the eyes versus a very slow alternation may reveal the differential effect of BS. Owing to the dynamic nature of BS, it may depend on the equipment utilized to control the monocular and binocular presentations. Consistent with our hypothesis, we discovered that BS is dynamic for the collinear configuration at close distances (2λ), and it relies on the stimulus presentation order, which was either mixed (BS exists) or non-mixed (BS is absent) between eyes. These results are consistent with our previous study. However, contrary to our previous study [110], here, we found that the effect of binocular collinear suppression was absent under the collinear 2λ condition at 200 ms; hence, BS existed when the stimulus presentation order was mixed between eyes.

## 2. Materials and Methods

### 2.1. Research Design

Our research design and hypothesis were taken from our recent lab studies [12,13,94,106,110] and mainly dealt with our previous study [110], specifically highlighting the difference in BS between the equipment utilized to control the monocular and binocular presentations, emphasizing the role of context. We tested BS under various spatial conditions: collinear (2λ, 3λ) and orthogonal (2λ, 3λ) for the control. Given the dynamic nature of BS for collinear 2λ, we conducted experiment 1(A-B) (see Table 1 and Figure 1 in the Section 2.2 for more details) to explore the influence of the testing order of the stimulus condition and the equipment utilized to control the monocular and binocular presentations. Consequently, we tested the collinear 2λ condition at 4 different presentation times (200, 120, 80, and 40 ms) following a gradual order of presentation times from longer to shorter durations.

We explored how ordered and non-ordered testing between the eyes (e.g., mixed and random) influences the BS. This may yield insights into the potential impact of decision criteria on BS. Here, we repeated the order of experiment 1 as in our previous study [110], following a gradual order of presentation times from longer to shorter ones to compare the results between the two studies, utilizing different BS testing methods.

### 2.2. Binocular Testing

We utilized 3D-Vision-2 Wireless Glasses (stereo glasses) to control the monocular and binocular presentations, which were utilized in previous studies [12,13,114]. The consumer version of NVIDIA 3D Vision consists of wireless LCD shuttered glasses that receive an infrared signal from an emitter connected to a PC via a USB cable. The glasses are shuttered at 120 Hz frequency; each eye was updated 60 times per second (60 Hz) for a flicker-free stereoscopic experience. An active shutter 3D system involves a technique of displaying stereoscopic 3D images. No cross talk was caused when utilizing the stereoscopic glasses.

By using this approach, the participants were not conscious of the eye whose image was presented. The Gabor patches were adjusted from a background luminance of 40 cd/m^2^, which was measured when utilizing the stereoscopic glasses.

#### 2.2.1. Experiment 1A

Across trials, the eyes (right, left, and binocular) were displayed randomly and by mixed trials. The order of stimulation across trials was intermixed at random for the eye presentation. Each eye’s stimulus was presented with a rapid alternation. For a monocular presentation, one eye’s stimulus was alternated with a mean luminance screen to the other eye, at a rate of 120 Hz. For binocular presentations (dichoptic presentation), the right and left eyes’ stimuli were alternated. By utilizing the “mixed between eyes” procedure, we mean that across trials, the eyes (right, left, and binocular) were displayed randomly and using mixed trials.

#### 2.2.2. Experiment 1B

Across trials, the eyes (right, left, and binocular) were displayed separately and by non-mixed trials. The order of the stimulation across trials was blocked for the eye presentation.

The purpose of our study was to explore the impact of the testing order of the stimulus condition on the BS. There was a notable difference between the two methods used: one involved a mixed (simultaneous) approach, while the other employed a non-mixed (sequential) approach.

#### 2.2.3. Comparation Between the Experiments

In experiment 1(A–B), we explored the impact of altering a single parameter, namely, the temporal aspect, the testing order of the stimulus condition, or the spatial distance on BS, while holding the other variables constant. To delve into the temporal aspect of binocular interactions and BS, we conducted tests utilizing 4 different presentation times: 200, 120, 80, and 40 ms. The order and sequence of these presentation times were treated as variables in each experiment (refer to Table 1 for more details). Another parameter under consideration was the spatial configuration, involving collinear configuration at 2λ and 3λ, as well as the orthogonal configuration at 2λ and 3λ.

### 2.3. Participants

The experiments, which took place at Bar-Ilan University, involved a total of 10 healthy naïve participants with ages ranging between 18 and 30 years old (27.5 ± 4.76 years, mean ± STD), who had either normal or corrected-to-normal vision. Only participants who passed a full optometric eye exam performed by an authorized optometrist were included (refer to Appendix A section for more details). The participants had healthy eyes, a visual acuity of 6/6 (Log-Mar 0) or better in each eye, and no more than one-line difference between eyes, whose eyes were fully corrected with no ocular disease or major phoria or amblyopia.

Prior to their participation, the participants provided their consent by signing a consent form that had been approved by the Internal Review Board (IRB) of Bar-Ilan University. This ensured compliance with the guidelines and regulations for human subject research. All experimental protocols were carried out according to the guidelines provided by the committee approving the experiments. All participants received financial compensation for their participation. Participants were enrolled by utilizing electronic advertisements and direct recruitment.

### 2.4. Apparatus

To present the stimuli, we utilized a windows PC computer connected to an LCD monitor (ASUS VG248QE, Taiwan, China) with a refresh rate of 120 Hz, controlled by a graphical card (NVIDIA GeForce GT730, Santa Clara, CA, USA), with a screen resolution of 1920 × 1080 pixels. Custom software (PSY, Bonneh, 2004, 2021d) was utilized for the experimental setup. The effective size of the screen measured 52 × 30 cm, which, when viewed from a distance of 150 cm, subtended a visual angle of 29° × 17°; additionally, gamma correction was applied.

### 2.5. The Stimuli

The stimuli were presented as gray-level images (Gabor patches, GPs) with an orientation of 90 degrees (vertical), a spatial frequency (SF) of 8 cycles per degree (cpd) with an equal wavelength (λ), and standard deviation (sigma, σ) (λ = σ = 0.21°), allowing for a minimum of 2 cycles in the GP. The stimuli were presented at 4 different presentation times, following a gradual order from the longest to shortest: 200, 120, 80, and 40 ms. The flanker’s contrast was 60% for presentation times of 200, 120, and 80 ms, and 90% for a presentation time of 40 ms. Note that, in our data, the flanker’s contrast (60 or 90%) that we utilized was 6-times above the target’s contrast detection threshold (10 or 15%), which may shift the facilitation to suppression [115]. In addition, it has been suggested [93,116] that suppression increases with increasing contrast and that the contrast detection threshold increases with decreasing presentation time. Thus, we would expect a higher contrast detection threshold for a presentation time of 40 ms [93,110,116,117] and more lateral suppression. To deal with this issue, we tried to keep the flanker-to-target contrast ratio similar by increasing the flanker’s contrast for a presentation time of 40 ms from 60% to 90%. The target–flanker orientation differences were either 0° (collinear configuration) or 90° (orthogonal configuration) with target–flanker separations of 2 or 3λ. The size of the stimuli for target–flanker separations of 3λ (center-center) subtends a visual angle of about 1.67° in the central visual field. Note that each stimulus display included four peripheral high-contrast crosses, marking the interval presentation of the target stimulus to avoid any uncertainty (see Figure 2). The stimulus contrast is defined by the Michelson formulation: I max − I min/I max + I min. The maximum (peak) possible contrast is defined as 127 luminance level (100%). All the Gabor patches were in the same phase, as in Figure 2 below.

In the current study, we utilized vertical Gabor stimuli following previous studies [13,111,114,117], since our lab’s prior work [111,112] reported abnormal and asymmetric monocular and binocular lateral interactions, specifically at the horizontal meridian for individuals with binocular fusion disorders (horizontal phoria). These studies revealed the absence of collinear facilitation at 3λ, specifically at the horizontal meridian. Consequently, these individuals exhibited a larger binocular perceptive field size, specifically at the horizontal meridian. Therefore, we believe that this procedure will yield more reliable data, since in the current study, we compared the contrast detection thresholds for both presentations monocularly and binocularly.

In the current study, we utilized Gabor patches (GPs) of 8 cycles per degree (cpd) based on our lab’s prior work [111,112] that found that the contrast detection thresholds for a single GP of 4 cpd were low. Therefore, we utilized a spatial frequency of 8 cpd for the GPs in the current study to prevent any potential floor effect of facilitation.

By utilizing the term “stimulus condition”, we mean that each condition involves subtypes of conditions, namely, spatial, temporal, or the presentation order.

### 2.6. Procedure

The two-temporal alternative forced-choice paradigm (2TAFC) and staircase procedure were utilized to measure the target contrast detection threshold, by utilizing a 3:1 staircase procedure, known to converge to a 79% correct response [118]. We examined the responses for a single target and a target with flankers (the LM paradigm) for both orthogonal and collinear configurations. Before each trial, a visible fixation circle was presented in the center of the screen; however, it disappeared when the trial was started by the participants at their own pace, by pressing the middle mouse button. During each trial, two stimuli were presented sequentially in a random order; only one had a target (two-temporal alternative forced-choice paradigm (2TAFC)). The participants’ task was to determine which stimuli included the target by pressing the left or right mouse keys (left for the first interval and right for the second). Visual feedback was provided for incorrect response after each presentation throughout the experiment.

Contrast detection thresholds were measured for an isolated single target or for flanked targets at target–flanker separations of 2 or 3λ in either orthogonal or collinear configurations for both monocular and binocular presentations. The spatial frequency of the target and flanker was 8 cpd. The stimuli were presented at 4 different presentation times, 200, 120, 80, and 40 ms, following a gradual order of presentation time from the longest to the shortest in each experiment (refer to Table 1 in the Section 2.2 for more details). We displayed a practice run utilizing a single target condition at presentation time of 200 ms, before the participants started the experiment.

Each experiment included 5 blocks (single target, collinear, and orthogonal at 2 and 3λ for each configuration). About 40 trials were needed to measure the contrast detection threshold in each block and for each condition (about 200 trials per session) for each presentation time. Each staircase (each data point) was repeated 3 times, for each eye condition (right, left, and binocular). So, the total number of trials for each participant was about 7200 trials. The data collection of the experiments lasted for 12–16 h, divided by an average of 2 h per day for 6–8 different days. On each day, the experiment was displayed with a different presentation time.

The BS was measured as previously calculated in the literature [2,12,13,111,114,117] as the ratio between the contrast detection thresholds of the average of two monocular eyes to the binocular contrast detection threshold. The BS ratio is the monocular/binocular contrast detection thresholds for each condition (single target, collinear, and orthogonal at 2 and 3λ for each configuration) at 4 different presentation times (200, 120, 80, and 40 ms). We found that there was no significant difference in the contrast detection threshold between the eyes for each condition during each presentation time during the different experiments, indicating that the performance of both eyes was similar. So, “monocular” in this study refers to the mean monocular (see Appendix A section for statistical information).

The threshold elevation (facilitation or suppression, collinear and orthogonal) was calculated for each eye condition (right, left, and binocular) at 4 different presentation times (at 2 and 3λ) as the log of the ratio between the masked target contrast detection threshold and the single target contrast detection threshold [threshold elevation = log (masked target contrast detection threshold/single target contrast detection threshold)]. Next, we compared the threshold elevation between the monocular and binocular presentations for collinear configurations at 2 and 3λ at 4 different presentation times. The specific tasks, conditions, and other details are listed in Figure 1 and Table 1 (refer to Section 2.2 for more details).

### 2.7. Data and Statistical Analysis

We performed one-way, two-way, and three-way mixed ANOVA to examine the impact of 1, 2, or 3 nominal variables (namely, presentation time, eye condition, and stimulus condition) on continuous outcomes, including contrast detection threshold, threshold elevation, or the BS ratio. Specifically, linear mixed effect models were performed, and the ANOVA was performed on the resulting models. All nominal variables were considered as fixed effects, and the participant’s ID was treated as a random effect. All interactions were involved in the initial models; however, if the interactions were non-significant, we refitted the models without these interactions. Post hoc analysis was conducted as pairwise comparisons defined by linear contrasts, and Benjamini–Hochberg (FDR) correction was applied to control for multiple testing. If the interactions were removed, we conducted the post hoc analysis by averaging the non-interacting factors. In cases when the outcome variable was a ratio, a logarithm transformation (with base 2) was applied. The normality of residuals and the homogeneity of variance assumptions were assessed graphically utilizing diagnostic plots. All data points were approved as not being outliers. All analyses were performed utilizing the R statistical environment (R Core Team (2021). R: A language and environment for statistical computing: the R Foundation for Statistical Computing, Vienna, Austria. URL https://www.R-project.org/ (accessed on 15 December 2021)).

## 3. Results

We explored the influence of spatial and temporal factors on the BS phenomenon utilizing the LM paradigm [90], highlighting both monocular and binocular interactions under various spatial and temporal conditions by utilizing stereo glasses to control the monocular and binocular presentations.

### 3.1. Experiment 1A: Longer to Shorter Presentation Times (Mixed Between Eyes)

In our previous study [110], we utilized mono-optic glasses and a sequential method (ordered). We found binocular suppression for the collinear 2λ condition at a presentation time of 200 ms (BS was absent), which was the first presentation time of the stimuli. However, the collinear suppression at a target–flanker separation of 2λ for the binocular presentation decreased after further presentations with different presentation times. Therefore, here, we utilized 3D-Vision-2 Wireless Glasses (NVIDIA) to control the monocular and binocular presentations (a simultaneous method). Across trials, the eyes (right, left, and binocular) were displayed randomly and by mixed trials. In this procedure, the observers were unaware of the eye whose image was presented. Our study aimed to explore how the testing order of the stimulus condition influences the BS phenomenon by utilizing a simultaneous method (mixed between eyes).

#### 3.1.1. Monocular vs. Binocular Contrast Detection Threshold

First, we measured the monocular and binocular contrast detection thresholds for a single target (isolated stimuli) and the target using the LM paradigm (stimuli with context). We utilized four presentation times in gradual order, from the longest to the shortest. Figure 3 presents the contrast detection threshold of a Gabor target, using monocular and binocular presentations. As expected, under both single target and collinear 3λ conditions, higher contrast detection thresholds were found for the shorter presentation time, whereas a lower contrast detection threshold was found for the longer presentation time using both monocular and binocular presentations. We observed a decrease in the contrast detection threshold (improvement) as a function of increasing presentation time. Hence, there was an improvement in the contrast detection thresholds by a factor of 1.4 using both monocular and binocular presentations with increasing presentation time; however, it reached a saturation at 120 and 200 ms. The effect of the presentation time was consistent with the literature [93,116] and consistent with our previous study. Interestingly, this effect was absent under the collinear 2λ condition; in other words, there was no improvement in the contrast detection threshold under the collinear 2λ condition in both monocular and binocular presentations as a function of increasing presentation time, which was consistent with our previous study (see Appendix A section). In summary, consistent with previous studies [93,116,117], when we investigated how the presentation time affects the contrast detection threshold, we found a decrease (improvement) in the contrast detection threshold with longer presentation times.

We utilized a three-way ANOVA test to examine the effect of the presentation time, the stimulus condition, and the group (the monocular or binocular presentation) on the contrast detection threshold. There was a significant effect of the presentation time (F (3,156) = 243.31, *p* = 0.00), the stimulus condition (F (4,156) = 54.43, *p* = 0.00), and the group [the monocular or binocular presentation] (F (1,156) = 355.45, *p* = 0.00) on the contrast detection threshold. Specifically, there was a significant difference between the monocular and binocular contrast detection thresholds under the single target condition at presentation times of 200,120, 80, and 40 ms (*p* = 0.00, *p* = 0.00, *p* = 0.00, *p* = 0.00 using Tukey’s post hoc analysis, respectively), which could be attributed to the BS effect. In addition, there was a significant interaction between the effect of the presentation time and the stimuli condition (F (12,156) = 3.78, *p* = 0.00). A significant interaction was observed between the effect of the presentation time and the group (F (3,156) = 6.59, *p* = 0.0003). In addition, there was a significant interaction between the effect of the stimuli condition and the group (F (4,156) = 9.29, *p* = 0.00). However, there was no significant interaction between the effect of the presentation time, the stimulus condition, and the group (F (12,156) = 0.83, *p* = 0.61).

#### 3.1.2. Monocular vs. Binocular Lateral Interactions

We compared monocular and binocular interactions using the collinear configuration for all presentation times at target–flanker separations of 2 and 3λ. Monocular and binocular facilitations were equal for all presentation times under the collinear 3λ condition (see Figure 4), which was consistent with a previous study [114]. However, under the collinear 2λ condition, there was no effect of binocular suppression or facilitation (see Figure 4) at longer presentation times (120 and 200 ms). Interestingly, during the shorter presentation times (40 and 80 ms), binocular collinear facilitation was observed only at shorter presentation times under the collinear 2λ condition (see Figure 4). There was a significant difference between the monocular and binocular presentations under the collinear 2λ condition (*p* = 0.00, using Tukey’s post hoc analysis after two-way ANOVA), whose monocular collinear facilitation was found under the collinear 2λ condition for all presentation times; the amount of monocular collinear facilitation was higher at the shorter presentation times (see Figure 4). Our results are consistent with and extend those in the literature [90,94,106,119], showing that facilitation existed for collinear configurations at target–flanker separations of 3λ, but they differed at 2λ. Contrast detection thresholds may be elevated (suppression) for shorter target–flanker separations, but in the current study, we found binocular collinear facilitation under the collinear 2λ condition, depending on the presentation time (see Figure 4).

When the stimulus presentation time was presented gradually from the longest to the shortest (200, 120, 80, and 40 ms), contrary to our previous study when we utilized mono-optic glasses to control the monocular and binocular presentations, we found a strong binocular collinear suppression at 200 ms. Here, utilizing stereo glasses to control the monocular and binocular presentations, we found that the effect of binocular collinear suppression was absent under the collinear 2λ condition at 200 ms. Thus, the results indicate that the method for testing the order of the stimulus condition [mixed between eyes in the current study vs. the sequential order can yield different results] (see Figure 4, Figure 5, Figure 6, Figure 7, Figure 8 and Figure 9).

A two-way ANOVA test was utilized to examine the effect of the presentation time and the stimulus condition on the threshold elevation. There was a significant effect of the presentation time (F (3,60) = 9.65, *p* = 0.00) and the stimulus condition (F (3,60) = 16.34, *p* = 0.00) on the threshold elevation. However, there was no significant interaction between the effect of the presentation time and the stimulus condition (F (9,60) = 1.09, *p* = 0.37).

#### 3.1.3. BS Phenomenon

The BS results are presented in Figure 5. Here, we found the BS effect for all presentation times and under all conditions and also under the collinear 2λ condition. Thus, note that, contrary to our previous study [110], here, when we displayed the experiment with the “mixed between eyes” (simultaneous), BS existed under the collinear 2λ condition.

**Figure 5 brainsci-14-01205-f005:**
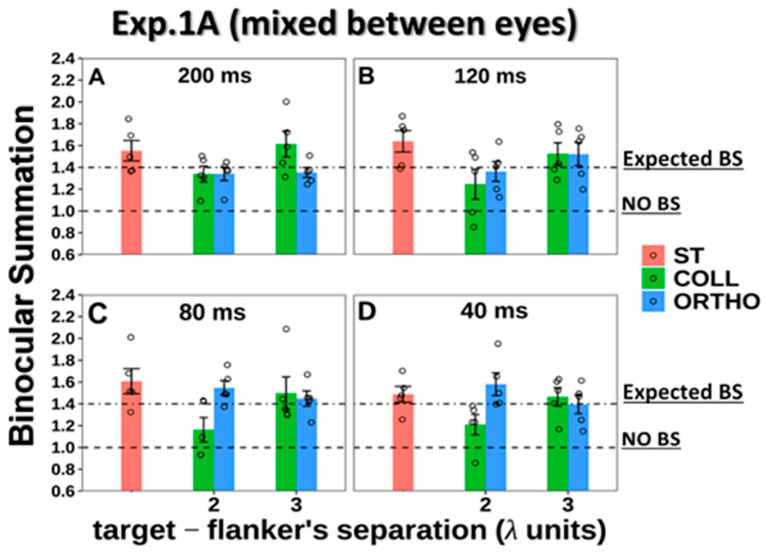
Binocular summation factor (monocular/binocular contrast detection threshold ratio) according to presentation times of 200, 120, 80, and 40 (ms) using the LM paradigm. The single target (ST), collinear configuration (COLL), orthogonal configuration (ORTHO) with target_flanker separations of 2 and 3λ for each configuration. (**A**) 200 (ms). (**B**) 120 (ms). (**C**) 80 (ms). (**D**) 40 (ms). N = 5, Error bars represent the standard error of the mean (SEM). Each dot represents an individual participant. The 1.4 dashed line represents the expected binocular summation (BS), whereas the 1 dashed line represents the absence of a BS effect.

A two-way ANOVA test was utilized to examine the effect of the presentation time and the stimulus condition on BS. We found a significant effect of the stimuli condition on BS (F (4,80) = 7.67, *p* = 0.00); however, no significant effect of presentation time on BS was found (F (3,80) = 0.05, *p* = 0.98), nor a significant interaction between the presentation time and the stimulus condition (F (12,80) = 0.89, *p* = 0.56).

Since there was no significant effect of the presentation time on BS (F (3,80) = 0.05, *p* = 0.98, two-way ANOVA), these conditions represented the mean values of the BS ratio (they were averaged) for all the presentation times under each stimulus condition (see Figure 6 below).

**Figure 6 brainsci-14-01205-f006:**
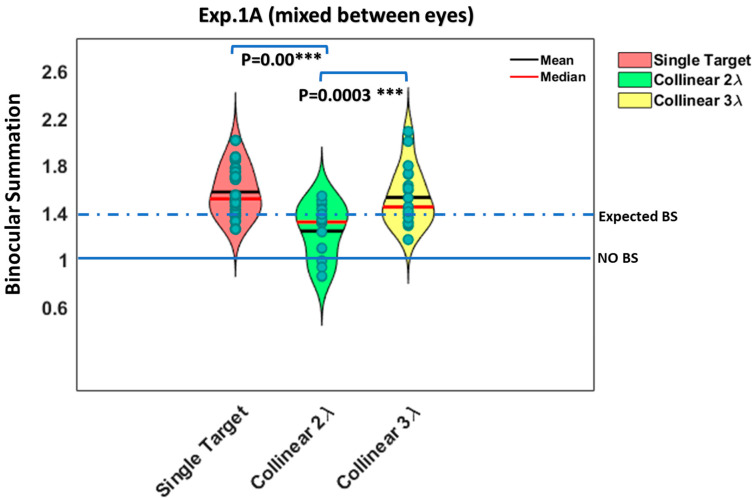
The violin plot shows the ratio of the monocular to binocular contrast detection threshold for 5 different participants averaged from experiment 1.A (mixed between eyes) for isolated stimuli vs. stimuli with context at 4 different presentation times (N = 5). We averaged all conditions as a mean value of the BS ratio for the different presentation times under each stimulus condition. We found that the binocular advantage was significantly greater for isolated (single target) than for closely flanked stimuli, collinear 2λ (*p* = 0.00 ***, using Tukey’s post hoc analysis after 2-way ANOVA). Each dot represents an individual participant. The 1.4 dashed line represents the expected binocular summation (BS), whereas the 1 solid line represents the absence of a BS effect.

To test the effect of spatial configuration on the BS, we performed the control condition by measuring the effect with the orthogonal configuration at target–flanker separations of 2 and 3λ for each experiment (see the results for the orthogonal configuration in Figure 5 in the Section 3 and Appendix A section). We found that under these conditions, the results were similar to those for the single target, indicating that BS existed and that there was no significant difference between single target and orthogonal configurations at target–flanker separations of 2 and 3 λ (*p* = 0.52, *p* = 0.31, using Tukey’s post hoc analysis after two-way ANOVA, respectively). Therefore, in the results of the following experiments, for simplicity, we presented the results only for the collinear configuration in the figures.

Similarly, the results of collinear 3λ were uniform for each experiment, showing collinear facilitation for both monocular and binocular presentations as well as BS similar to a single target, which was consistent with previous studies [13,111]. However, we continue to present this information, since it was found in our previous study and others [94,106,113] that interactions at collinear 3λ affects the collinear interactions at 2λ. In the following experiment, we aimed to investigate how the testing order of the stimulus condition affects the binocular interactions and BS.

### 3.2. Experiment 1B: Longer to Shorter Presentation Times (Non-Mixed Between Eyes)

In experiment 1.A (mixed between eyes), the stimulus presentation time order during the experiment was presented gradually from the longest to the shortest (200,120, 80, and 40 ms), utilizing stereo glasses to control the monocular and binocular presentations. The results indicated that the effect of binocular collinear suppression was absent under the collinear 2λ condition at 200 ms. This result differs from the result when we utilized mono-optic glasses to control the monocular and binocular presentations [110], which showed a strong binocular collinear suppression at 200 ms. In addition, in experiment 1A (mixed between eyes), we found binocular collinear facilitation for the collinear 2λ condition, depending on the presentation time (see Figure 4). These results might be attributed to the influence of the testing order of the stimulus condition and by perceptual practice. Therefore, we performed experiment 1B (non-mixed between eyes) to compare the two studies and to explore how the testing order of the stimulus condition (non-mixed between eyes) influences the binocular interactions and BS.

First, we measured the contrast detection thresholds of single targets (isolated stimuli) and targets using the LM paradigm (stimuli with context) according to four different presentation times following a gradual order, from the longest to the shortest, using monocular and binocular presentations (see Appendix A section).

#### 3.2.1. Monocular vs. Binocular Lateral Interactions

We compared monocular and binocular interactions using a collinear configuration for all presentation times at target–flanker separations of 2 and 3λ. For the binocular presentation, detection was facilitated by collinear flankers at target–flanker separations of 3λ; however, it was strongly suppressed under the collinear 2λ condition (see Figure 7A). These results were consistent with previous studies and extended them [13,90,111]. Interestingly, the monocular and binocular facilitations were equal under the collinear 3λ condition. However, under the collinear 2λ condition, binocular suppression was found (see Figure 7A; however, the binocular suppression was significantly higher than the monocular facilitation; (*p* = 0.00, using Tukey’s post hoc analysis after 2-way ANOVA)). There was a significant effect of the stimulus condition (F (3,60) = 44.50, *p* = 0.00, two-way ANOVA) on the threshold elevation. However, there was no significant effect of the presentation time (F (3,60) = 2.10 *p* = 0.11, two-way ANOVA). In addition, there was no significant interaction between the effect of the presentation time and the stimuli condition (F (9,60) = 1.57, *p* = 0.14, two-way ANOVA). Specifically, there was a significant difference between the monocular and binocular presentations under the collinear 2λ condition (*p* = 0.00, using Tukey’s post hoc analysis after 2-way ANOVA).

**Figure 7 brainsci-14-01205-f007:**
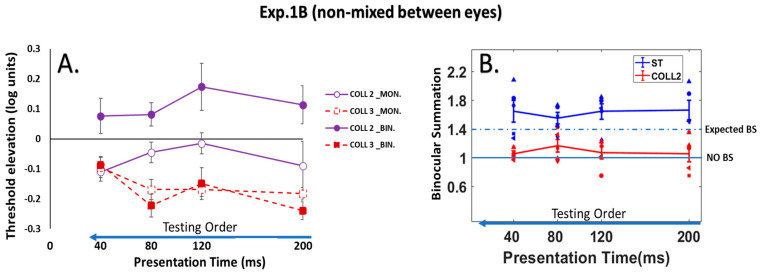
The order of the stimulation across trials was blocked for the eye presentation. (**A**) Collinear Interactions as a function of the presentation time (200, 120, 80, and 40 ms). Monocular (MON.), binocular (BIN.), collinear configuration (COLL) with target_flanker separations of 2 and 3λ. Facilitation is indicated by values below zero, and suppression by values above zero. N = 5. Error bars represent the standard error of the mean (SEM). The blue arrow points to the left from longer to shorter presentation times, which describes the testing order of the stimulus condition. The solid purple lines represent the collinear interactions at 2λ, denoted by open and filled circles for monocular and binocular presentations, respectively, whereas the dashed dark red lines represent the collinear interactions at 3λ, denoted by open and filled squares for monocular and binocular presentations, respectively. (**B**) The line plot represents the distribution of the binocular summation factor (the monocular/binocular contrast detection threshold ratio) as a function of the presentation time (200, 120, 80, and 40 ms) utilizing the LM paradigm for 5 different participants (each different shape represents an individual participant). Single target (ST), collinear configuration with target_flanker separations of 2λ (COLL2). N = 5, error bars represent the standard error of the mean (SEM). The 1.4 dashed line represents the expected binocular summation (BS), whereas the 1 solid line represents the absence of a BS effect.

#### 3.2.2. BS Phenomenon

The BS results are presented in Figure 7B. Here, we found the opposite effect compared to the results when the stimulus presentation order was mixed between eyes (experiment 1A). We found BS under the single target, collinear 3λ, and orthogonal 2 and 3λ conditions at all presentation times (see the results for the orthogonal configuration in Appendix A section). However, no BS was found under the collinear 2λ condition at all presentation times; here, we found suppression under the collinear 2λ condition for all presentation times with the binocular presentation. Interestingly, when the stimulus presentation order was non-mixed between eyes (sequential), there was no BS for collinear 2λ, suggesting that the order affects the binocular interactions under the collinear 2λ condition, which affects BS.

A two-way ANOVA test was utilized to examine the effect of the presentation time and the stimulus condition on BS. Interestingly, we found a significant effect of the stimulus condition on BS (F (4,76) = 25.73, *p* = 0.00); however, no significant effect of the presentation time on BS was found (F (3,76) = 1.74, *p* = 0.16), nor a significant interaction between the presentation time and the stimulus condition (F (12,76) = 0.79, *p* = 0.65). Since there was no significant effect of the presentation time on BS (F (3,76) = 1.74, *p* = 0.16, two-way ANOVA), these conditions represented the mean values of BS (they were averaged) for all presentation times under each stimulus condition (see Figure 8).

**Figure 8 brainsci-14-01205-f008:**
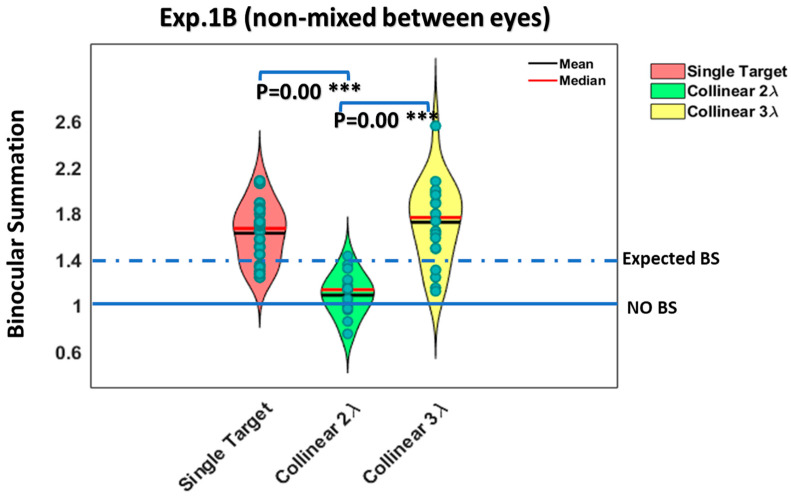
The violin plot shows the ratio of the monocular to binocular contrast detection threshold for 5 different participants averaged from experiment 1.B (non-mixed between eyes) for isolated stimuli vs. stimuli with context at 4 different presentation times (N = 5). We averaged all conditions as a mean value of the BS ratio for all the presentation times under each stimulus condition. We found that the binocular advantage was significantly greater for isolated (single target) than for closely flanked stimuli, collinear 2λ (*p* = 0.00 ***, using Tukey’s post hoc analysis after 2-way ANOVA). Each dot represents an individual participant. The 1.4 dashed line represents the expected binocular summation (BS), whereas the 1 solid line represents the absence of a BS effect.

Importantly, initially, it seems that the collinear 3λ data overlap for every subject except one at the very top of the lollipop in Figure 8, which may suggest that this subject explains the difference for the collinear 3λ condition, rather than there being a systematic shift in the distribution. We checked this issue in the BS ratio data for the collinear 3λ condition in Figure 8. We found that the mean was 1.72 and the median was 1.76 (including the participant with the high ratio of BS = 2.55), whereas when we calculated the mean and the median without including this subject, we found that the mean was 1.67 and the median was 1.73. In addition, we performed a Welch two-sample *t*-test with equal variance (paired Student’s *t*-test) to evaluate the statistical differences between the two datasets for the collinear 3λ condition in Figure 8 (including this subject and without including this subject). We found that there was no significant difference between the two datasets (*p* = 0.68, paired, two-tailed, *t*-test). Therefore, we decided to keep and include this subject in the data. Furthermore, the normality of residuals and the homogeneity of the variance assumptions were assessed graphically utilizing diagnostic plots; in our study, all data points were approved as not being outliers (see the Section 2.7 in Section 2).

### 3.3. Summary Across the Different Experiments (Exp.1 A-B)

Figure 6 and Figure 8 present BS for two different experiments under each stimulus condition (they were averaged) for all four different presentation times. Since there was no significant effect of the presentation time on BS for each experiment (*p* = 0.98, *p* = 0.16, two-way ANOVA), we averaged all conditions as a mean value of BS for all presentation times, for each stimulus condition, and for each experiment (see Figure 6 and Figure 8).

Figure 6 presents the BS for experiment (1A) utilizing a simultaneous method (mixed between eyes) for each stimulus condition (they were averaged) for all four different presentation times. Figure 8 presents the BS for experiment (1B) utilizing the sequential method (non-mixed between eyes) for each stimulus condition (they were averaged) for all four different presentation times.

Next, we utilized a two-way ANOVA test to examine the difference between the two sets of experiments and the stimulus conditions for BS. No significant effect of the experiment on BS was found (F (1,15.06) = 2.35, *p* = 0.15, two-way ANOVA); however, we found a significant effect of the stimulus condition on BS (F (2,102.86) = 47.14, *p* = 0.00, two-way ANOVA), and a significant interaction was found between the stimulus condition and the experiment (F (2,102.86) = 5.39, *p* = 0.006, two-way ANOVA). Next, we utilized a post hoc analysis for the significant main effect of the stimulus condition. Specifically, there was a significant difference between experiment (1A) (mixed between eyes) and experiment (1B) (non-mixed between eyes) under the collinear 3λ condition (*p* = 0.01, using Tukey’s post hoc analysis after 2-way ANOVA). However, there was no significant difference between the experiments for single targets or under collinear 2λ conditions (*p* = 0.30, *p* = 0.43, using Tukey’s post hoc analysis after 2-way ANOVA, respectively).

In summary, under the collinear 2λ condition, BS is not consistent; however, it relies on the testing order of the stimulus condition. BS is dynamic under the collinear 2λ condition, whereas under single target, collinear 3λ, and orthogonal 2 and 3λ conditions, BS is consistent. Thus, the testing order of the stimulus condition affects the binocular interactions and BS under the collinear 2λ condition when the stimulus presentation time order during the experiment is presented from longest to shortest with mixed (BS exists) or non-mixed (BS is absent) between eyes (see Figure 5, Figure 6, Figure 7 and Figure 8).

## 4. Discussion

The results of our previous study [110] raised a challenging question as to whether the participants were aware of shifting the testing between the eyes during the stimulus presentation, which may affect the BS mechanisms. Here, we repeated the main experiments utilizing stereo glasses, in which the observers were unaware of the eye that perceived the stimuli. Therefore, a comparison between the two studies may help to determine whether BS may be influenced by additional factors such as awareness. Alternatively, there may be an effect of long-term persistence of subthreshold lateral excitation from collinear 3λ, which can persist between the trials of different eyes; thus, any subthreshold lateral excitation could decrease the suppression at collinear 2λ [90,94,95,106,108,110,113], consequently affecting BS.

In the current study, we utilized stereo glasses to control the monocular and binocular presentations. We assumed that BS is not consistent; it relies on the equipment utilized to control the monocular and binocular presentations either by mono-optic glasses or stereo glasses. Consistent with our hypothesis, we found that BS is dynamic for the collinear configuration at 2λ, depending on the stimulus presentation order for the eye condition, which was either mixed (BS exists) or non-mixed (BS is absent) between eyes for the collinear but not for the orthogonal configuration. However, BS is uniform at more distant flankers (collinear and orthogonal, 3λ). Thus, BS is not uniform; it depends on the stimulus condition, the presentation times, the testing order of the stimulus condition, and the equipment utilized to control the monocular and binocular presentations either by mono-optic or stereo glasses.

Consistent with our previous study [110], we found that BS is not uniform, but, contrary to our previous study [110], here, we found that the effect of binocular collinear suppression was absent under the collinear 2λ condition at 200 ms; hence, BS existed, which could be attributed to the influence of the testing order of the stimulus condition (mixed between eyes) (see Figure 9).

A difference exists between the results under the collinear 2λ condition when utilizing different glasses to control the monocular and binocular presentations. When utilizing mono-optic glasses [110], the binocular collinear suppression was significantly higher than the monocular presentation at 200 ms. This effect was found when the order of the presentation time was presented gradually from the longest to the shortest (200, 120, 80, and 40 ms). However, the testing order of the stimulus condition affected the results; the collinear suppression decreased for both monocular and binocular presentations under the collinear 2λ condition as a function of the presentation time. Contrary to these results, in the current study, we used the same order of presentation time (200, 120, 80, and 40 ms), but we utilized stereo glasses (mixed between eyes) to control the monocular and binocular presentations. Interestingly, we found that the effect of the binocular collinear suppression was absent under the collinear 2λ condition at 200 ms (see Figure 9). When we compared the results between the two methods (mono-optic glasses vs. stereo glasses), we found that the binocular collinear suppression under the collinear 2λ condition at 200 ms was significantly higher (*p* = 0.0015, Welch two-sample *t*-test, un-paired) for the mono-optic glasses experiment than for the stereo glasses experiment (see Figure 9).

**Figure 9 brainsci-14-01205-f009:**
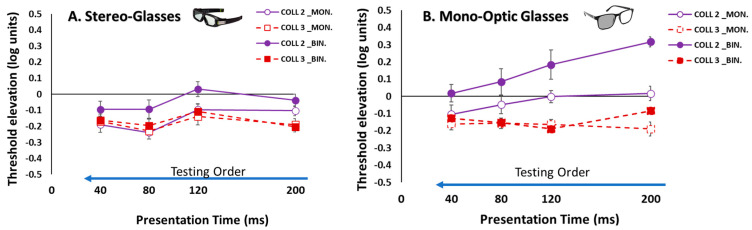
Collinear Interactions as a function of the presentation time (200, 120, 80, and 40 ms). Monocular (MON.), binocular (BIN.), collinear configuration (COLL) with target_flanker separations of 2 and 3λ. Facilitation is indicated by values below zero, and suppression by values above zero. N = 5. Error bars represent the standard error of the mean (SEM). The blue arrow, which points to the left from longer to shorter presentation times, describes the testing order of the stimulus condition. The solid purple lines represent the collinear interactions at 2λ, denoted by open and filled circles for monocular and binocular presentations, respectively, whereas the dashed dark red lines represent the collinear interactions at 3λ denoted by open and filled squares for monocular and binocular presentations, respectively. (**A**). The current study utilizes stereo glasses to control the monocular and binocular presentations. (**B**). Taken from our previous study (https://doi.org/10.1038/s41598-023-48380-2 (accessed on 5 December 2023)) utilizing mono-optic glasses to control the monocular and binocular presentations.

### 4.1. Comparison with Other BS Studies Utilizing Different Methods for Binocular Testing

Over the years, the computational models of BS and interactions have become quite complex. However, recent models revealed simpler explanations regarding the general effect of collinear 2λ. Meese, Challinor, and Summers [120] utilized shutter goggles for binocular testing and used superimposed pattern masks to interpret the loss of BS from masking (when the mask is outside the excitatory range of the detecting mechanism). They showed that when masking is absent, the observer benefits from signals in two eyes compared with one; however, when the mask contrast is high, a larger masking effect exists, and there is no binocular advantage. Another model by Lev et al. [13] included the gain control model; it utilized dichoptic glasses to control the monocular and binocular presentations and successfully captured the abolished BS by nearby flankers. Baker et al. (2018) [3] suggested that the best way to measure BS is by utilizing equipment, such as stereoscopes, shutter goggles, or virtual reality designed for a binocular presentation. An improvement of a factor of about 1.4 was found for isolated targets [17,18,19,69,70,71,72], suggesting a quadratic summation of the two monocular inputs (√2). However, a recent review reported that different amounts of BS exist in a range from √2 to 2 values.

On the other hand, studies that utilized visual evoked potential (VEP) [121] reported that BS can be above 2. Apkaria et al. demonstrated that BS can be expressed as values ranging from 1 to 5 or “to infinity” with neuronal responses. Furthermore, BS can be extended indefinitely in binocular cortical neurons; many of them show no response to the stimulation of monocular presentations.

The variability in the reported amount of BS may be due to the different equipment used to test BS. Models of BS have been elaborated for isolated stimuli [14,15,16,17,18,19,20,21], whereas other models of BS have been elaborated for stimuli with context [13,84,85,99,120,122,123,124,125,126,127,128,129,130,131,132]. These studies utilized different equipment to control the monocular and binocular presentations across different studies, using equipment, such as stereoscopes, frosted occluder, shutter goggles, mono-optic glasses, virtual reality, mirror stereoscope, dichoptic presentation, or patching the unstimulated eye (the latter may have been avoided, if possible, to prevent binocular rivalry).

In our study, the fact that BS under a certain context recovers at collinear 2λ, relying on the presentation order under the eyes’ condition, focusing on the possibility that lateral interactions of excitation from collinear 3λ, which reduce the suppression (the masking effect) at collinear 2λ, is more likely to explain our results. Lev et al. tested this issue, reporting that a gain control theory can possibly explain the absence of BS under the collinear 2λ condition; this strongly suggests that monocular interactions take place before the site of a binocular combination. Thus, lateral suppression may cancel or reduce the binocular facilitation. Thus, our suggestion is consistent with the computational models of Meese [120] and Lev [13].

### 4.2. Comparison Between Our Previous and Current Study

There was an important factor in the research design for determining the awareness of the participant whose eye was tested as well as the occlusion method (the stereo glasses utilized in the current study vs. mono-optic glasses, utilized in our previous study). There was a significant difference between the two methods. Here, when utilizing the stereo glasses, the participants were unaware of the eye whose image was presented. For the monocular presentation, one eye’s stimulus was alternated with a mean luminance screen to the other eye, at a rate of 60 Hz for each eye. For the binocular presentation, the right and left eye’s stimuli were alternated, at a rate of 120 Hz. The results indicate that, contrary to our previous study [110], here, we found that the effect of binocular collinear suppression was absent under the collinear 2λ condition at 200 ms (see Figure 9); hence, BS existed.

Our results can be explained by the dynamics of binocular interactions (suppression and/or facilitation): inter-ocular (between eyes) and local (monocular within each eye). The results suggest that BS combines both types of interactions, i.e., suppression and/or facilitation, depending on whether the target–flanker separations are either at 2 or 3λ.

Note that for the single target condition, 75% of the participants reached a BS ratio ≥ 1.4, whereas for 25% of them, the BS ratio was between 1 and 1.4 (1 ≤ BS < 1.4). However, under the collinear 2λ condition, only 17.5% of the participants reached a BS ratio ≥ 1.4 (see Figure 6 and Figure 8), whereas 52.5% of the participants reached a BS ratio between 1 and 1.4 (1 ≤ BS < 1.4). Interestingly, 30% of the participants reached a BS ratio < 1 (suppression), probably due to the inter-ocular suppression effect. Importantly, under the collinear 3λ condition, 72.5% of the participants reached a BS ratio ≥ 1.4, whereas for 27.5% of them, the BS ratio was between 1 and 1.4 (1 ≤ BS < 1.4). Importantly, 0% of the participants reached a BS ratio of <1, probably due to the absence of the inter-ocular suppression effect under the collinear 3λ condition (see Figure 6 and Figure 8). Our results indicate that under the collinear 2λ condition (but not under the orthogonal condition), detection is suppressed more under binocular than under monocular presentation, whereas more distant flankers (collinear 3λ) facilitate both monocular and binocular detection. We found that two eyes are not better than one with crowded targets (the collinear 2λ condition), which is consistent with and extended previous studies [12,13,110]. In other words, we found that a BS of the contrast detection threshold was absent at close distances (collinear 2λ), depending on the stimulus presentation order for the eye condition (mixed or non-mixed between eyes), for the collinear but not for the orthogonal configuration. However, a BS of the contrast detection threshold existed at more distant flankers (collinear and orthogonal, 3λ), which is consistent with and extended previous studies [13,111].

### 4.3. Monocular vs. Binocular Lateral Interactions

Our results did not reveal an advantage of binocular collinear facilitation for all presentation times, suggesting that monocular and binocular collinear facilitations were equal, which was consistent with and extended previous studies [13,111,114]. We found collinear facilitation at all presentation times at larger distances (collinear 3λ). For closer distances (collinear 2λ), the facilitation depends on the stimulus presentation order for the eye condition (mixed or non-mixed between eyes) and on the stimulus presentation time. Our results are consistent with those in the literature and extend them [90,94,106,119], showing that facilitation existed for collinear configurations at a target–flanker separation of 3λ, but they were different at 2λ; the contrast detection thresholds may be elevated (suppression) for shorter target–flanker separations.

We suggest that the dynamics of BS under the collinear 2λ condition can be explained by the Hebbian learning rules. It has been found [95,106,108,113] that collinear facilitation can persist up to a few minutes. Thus, by mixing trials of collinear 2 and 3λ, the facilitation from collinear 3λ can persist even when the conditions change; thus, facilitation can affect the trials of collinear 2λ. Therefore, we suggest that the dynamic effect of BS at close distances, i.e., collinear 2λ, can be explained by Hebbian activity-dependent synaptic changes (functional plasticity). Inputs that activate neurons within short times are mutually strengthened; they excite neighboring neurons in the primary visual cortex (V1) through long-range horizontal connections. In our study, we utilized the lateral masking paradigm in which the mechanisms for the collinear configuration can be explained as follows [90,94,95,106,108,113]: an excited neuron affects its neighboring neurons by inhibiting or exciting their activity; this produces a network of long-range connections that exist between similar orientation columns. Specifically, each mask stimulus activates its neighboring neurons, which are organized topographically. In this way, masks corresponding to neighboring locations in the visual field activate neighboring neurons. Polat and colleagues [106] found that signals that propagate through these connections are limited by certain time windows (approximately a few minutes) and that they propagate only if both neurons involved are activated (Hebbian learning). In other words, the long-range connections are created by a cascade of local interactions (a sequence of neural activations).

### 4.4. Dynamics of the Perceptive Field Size Under the Collinear 2λ Condition

We analyzed the results from our previous study [110] and the current study to estimate the perceptive field size (the fundamental processing unit of human vision, measured psychophysically by a perceptual response) [133]. Our studies [110,133] show that the order in which stimulus presentation times were displayed influences the binocular interactions and BS. Research findings indicate that under the collinear 2λ condition, BS is not consistent and uniform; however, it relies on the sequence in which the conditions are tested. This dynamic effect can be attributed to the reduced lateral suppression at collinear 2λ. So, BS is dynamic under the collinear 2λ condition due to the dynamics of lateral suppression. This effect leads to a decrease in the size of the perceptive field (PF). Moreover, it was reported that the lateral suppression decreased through repetitions during the experiment for both monocular and binocular presentations. This effect leads to a decrease in the PF size.

### 4.5. Comparison of Contrast Threshold Between Eyes at Different Presentation Times

BS implies that the binocular contrast detection threshold is better than the monocular contrast detection threshold, with an improvement of a factor of about 1.4 (empirically). For the collinear 2λ condition, this is true at the longer presentation times (200 and 120 ms), as shown in Figure 3 and Figure 5. However, this effect was reduced at the shorter presentation times (80 and 40 ms). The binocular contrast detection thresholds were almost equal to the monocular ones. This raised a challenging question: whether some of the subjects were seeing out of one eye only. It is true that the binocular response for short presentation times resembles those at the monocular level. This is also demonstrated in the study [13] of Lev et al., 2021. This is challenging to predict or determine from the monocular information, such as visual acuity regarding which eye contributes to the results. Since we have the visual acuity (VA) data for both eyes, we can make a correlation between VA and the contrast detection threshold of the Gabors that were presented at brief (shorter) presentation times. It is reasonable to expect that lower VA in one eye may lead to less visibility of the target in the weaker eye, as was shown in our previous study [117]. We checked this issue in the VA data and found that there was no difference between the right and left eyes regarding VA. The subjects had healthy eyes, a visual acuity of 6/6 (Log-Mar 0) or better in each eye, and there was no difference between eyes even one-line in individuals whose eyes were fully corrected and who had no ocular disease or major phoria or amblyopia (see the Section 2.3 in Section 2 and Appendix A section for more details). In addition, we found that there was no significant difference in the contrast detection threshold between the eyes for each condition during each presentation time during the different experiments (see the Section 2.6 in Section 2), indicating that the performance of both eyes was similar; therefore, “monocular” in this study refers to the mean monocular (see Appendix A section for statistical information).

### 4.6. There Might Be a Few Possible Views That Attempt to Explain the Origin of Inhibition Under the Collinear 2λ Condition

Usually, it is explored within the lateral masking experiment and discussed as an effect within the same receptive field or between different receptive fields based on animal physiology studies. Carandini et al. [134] proposed that many phenomena, such as contrast saturation and cross-orientation suppression, are attributed to cortical inhibition. However, they can be explained by thalamo-cortical synaptic depression rather than intra-cortical inhibitory mechanisms (the responses of neurons in the primary visual cortex (V1) were suppressed by mask stimuli that do not elicit responses if presented alone). The physiological evidence also supports a primary role for thalamo-cortical synaptic depression. This mechanism provides a rapid, localized, and monocular adjustment to the stimulus conditions, directly linking synaptic-level physiology to behavioral visual phenomena. However, support for the inhibition in V1, linking physiology and the psychophysical results, was provided in the studies of Polat and colleagues [91,100,104], showing that the flankers alone did not evoke responses from the RF but that adding collinear flankers facilitated or suppressed the target response. This result shows that the lateral excitation controls the gain in the RF.

### 4.7. Is BS Under the Collinear 2λ Condition Affected by Awareness?

To further address this question, we compared two procedures utilizing stereo glasses, in which the observers were unaware of the eye that perceived the stimuli: the mixed between eyes (experiment 1A) and the non-mixed between eyes (experiment 1B). We found different results for the two experiments. BS existed in the mixed between eyes procedure (experiment 1A), whereas BS was absent in the non-mixed between eyes procedure (experiment 1B) for the collinear 2λ condition. Therefore, BS under the collinear 2λ condition may be affected by awareness. If we obtained different results in the two experiments, does this suggest that awareness affects BS? Is it really awareness or the occlusion method itself? We suggest that it may also be affected by other factors such as the testing order of the stimulus condition using the procedure between the eyes, specifically, long-term persistence of subthreshold lateral excitation from collinear 3λ, which can persist between the trials of different eyes, and it decreases the suppression at collinear 2λ, consequently affecting BS.

## 5. Conclusions

Our findings revealed that BS under the collinear 2λ condition is not consistent; rather, it relies on various factors, such as the order in which the stimulus conditions were presented, either mixed (BS exists) or non-mixed (BS is absent) between eyes, as well as the presentation time and the equipment utilized to control the monocular and binocular presentations. The reason for these variations can be attributed to the dynamics of binocular interactions, including suppression and/or facilitation, specifically, between eyes (inter-ocular) and monocular within each eye (local), relying on whether the target–flanker separations are either at two or three wavelengths (λ). We suggest that for the collinear 2λ condition, the BS exhibited a dynamic characteristic, while it remained uniform under the single target, collinear 3λ, and orthogonal 2 and 3λ conditions.

## Figures and Tables

**Figure 1 brainsci-14-01205-f001:**
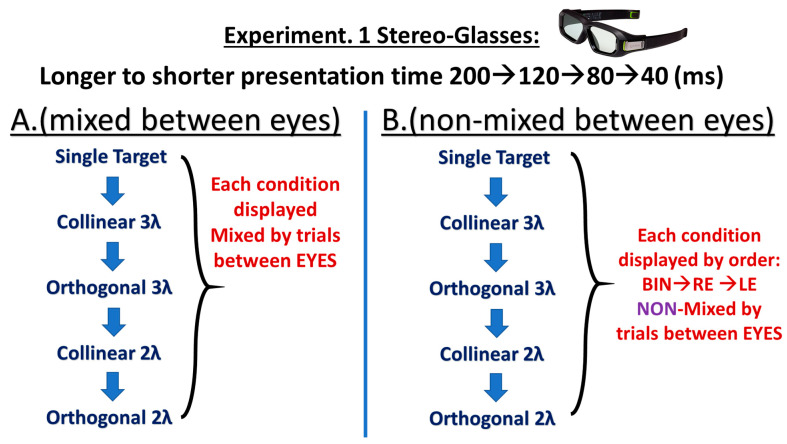
Details of the experimental design. The testing order of the stimulus condition for blocks and sessions are presented in the diagram. The BIN binocular, RE right eye, LE left eye, single target, collinear, and orthogonal configurations at target_flanker separations of 2 and 3λ for each configuration. The stimuli were displayed at 4 different presentation times: 200, 120, 80, and 40 ms following a gradual order from the longest to the shortest. All the conditions were displayed for one presentation time first, and then displayed for the next presentation time. The two-temporal alternative forced-choice paradigm (2TAFC) and the staircase procedure were utilized to measure the target contrast detection threshold. Stereoscopic glasses were utilized to control the monocular and binocular presentations. Each data point was repeated 3 times (A). Experiment 1.A (mixed between eyes): We used 5 files; each file included a different stimulus condition namely single target, collinear 3λ, orthogonal 3λ, collinear 2λ, and orthogonal 2λ. Each file included three blocks: the first block for the binocular condition (BIN), the second block for the right eye (RE), and the third block for the left eye (LE); Across trials, the eyes (right, left, and binocular) were displayed randomly, and by mixed trials. (B). Experiment 1.B (non-mixed between eyes): We used 3 files; the first file included one block for the single target condition, the second file included two blocks for collinear and orthogonal 3λ conditions, and the third file included two blocks for collinear and orthogonal 2λ conditions. For each stimulus condition we tested the binocular, right, and left eye, respectively. Across trials, the eyes (right, left, and binocular) were displayed separately, and by non-mixed trials.

**Figure 2 brainsci-14-01205-f002:**
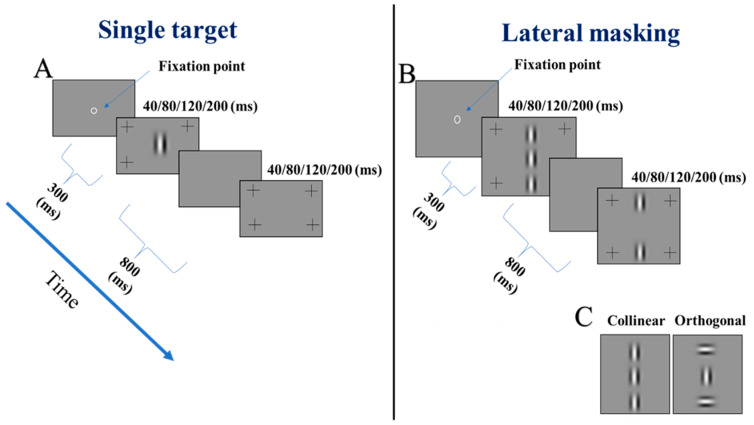
The lateral masking paradigm (LM). Stimuli that were utilized in the research. (**A**) Gabor target example was utilized in the experiments: single target condition. (**B**) The LM paradigm. (**C**) Spatial configurations that were utilized in the research: collinear (**left**) and orthogonal (**right**). To measure the target contrast detection threshold, the two-temporal alternative forced-choice paradigm (2TAFC) and a 3:1 staircase procedure, known to converge to a 79% correct response, were utilized. Participants were required to determine in which interval (the first or second) the central Gabor target has been presented as illustrated in panel (**B**).

**Figure 3 brainsci-14-01205-f003:**
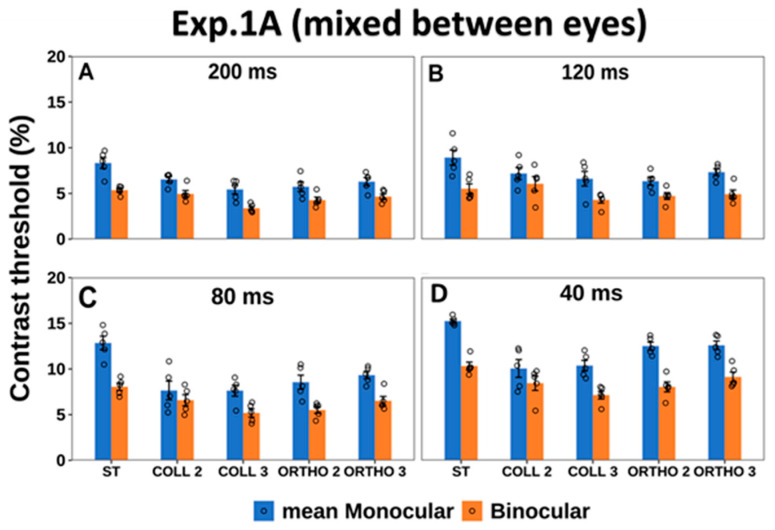
The mean monocular and binocular contrast detection thresholds according to the presentation times of 200, 120, 80, and 40 (ms) using the LM paradigm. The single target (ST), collinear configuration (COLL), and orthogonal configuration (ORTHO) with target_flanker separations of 2 and 3λ for each configuration. (**A**) 200 ms. (**B**) 120 ms. (**C**) 80 ms. (**D**) 40 ms. N = 5, Error bars represent the standard error of the mean (SEM). Each dot represents an individual participant.

**Figure 4 brainsci-14-01205-f004:**
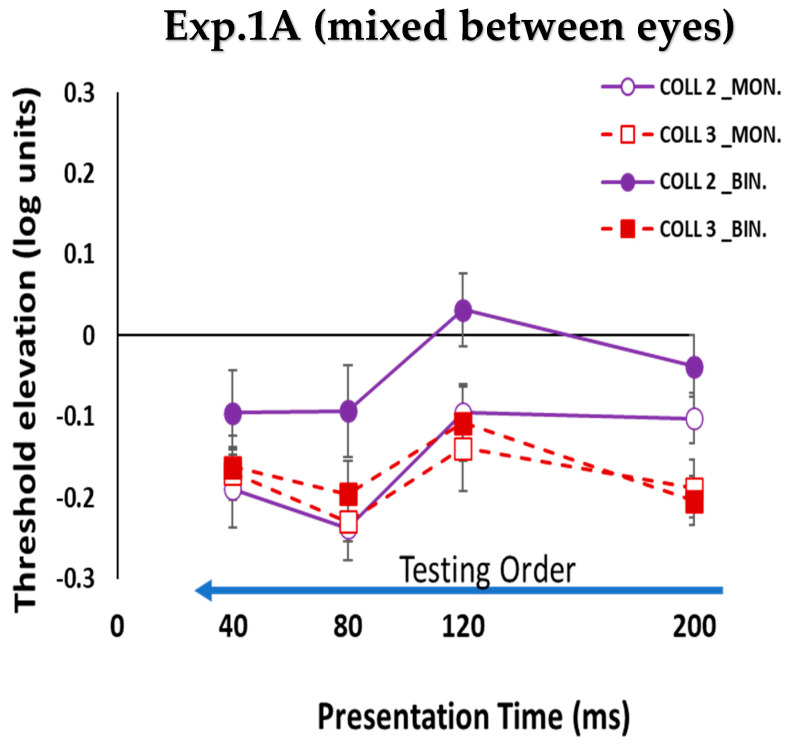
The order of the stimulation across trials was intermixed at random for the eye presentation. Collinear interactions as a function of the presentation time (200, 120, 80, and 40 ms). Monocular (MON.), binocular (BIN.), collinear configuration (COLL) with target_flanker separations of 2 and 3λ. Facilitation is indicated by values below zero, and suppression by values above zero. N = 5. Error bars represent the standard error of the mean (SEM). The blue arrow points to the left from longer to shorter presentation times that indicate the testing order of the stimulus condition. The solid purple lines represent the collinear interactions at 2λ, denoted by open and filled circles for monocular and binocular presentations, respectively, whereas the dashed dark red lines represent the collinear interactions at 3λ, denoted by open and filled squares for monocular and binocular presentations, respectively. [threshold elevation = log (masked target contrast detection threshold/single target contrast detection threshold)]. Next, we compared the threshold elevation between the monocular and binocular presentations for collinear configurations at 2 and 3λ at 4 different presentation times.

**Table 1 brainsci-14-01205-t001:** Details of the experimental design. The BIN binocular, RE right eye, LE left eye, ST single target, COLL collinear configuration, and ORTHO orthogonal configuration at target–flanker separations of 2 and 3λ for each configuration.

Experiment	Order of Presentation Time	Order of the Stimuli	Eye’s Condition
1.A	Longer to shorter (200 → 40 ms)	ST → COLL&ORTHO(3λ) → COLL&ORTHO(2λ)	Mixed procedure
1.B	Longer to shorter (200 → 40 ms)	ST → COLL&ORTHO(3λ) → COLL&ORTHO(2λ)	BIN → RE → LE

## Data Availability

The datasets used and/or analyzed during the current study are available from the corresponding author on reasonable request due to privacy restrictions.

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
