# Peer review of "What Factors Affect Binocular Summation?"

_brainsci, 2024, doi:10.3390/brainsci14121205_

Round 1

Reviewer 1 Report

Comments and Suggestions for Authors

The study investigates factors influencing binocular summation (BS) by comparing results from mono-optic and stereo-glasses presentations. It reveals that BS varies depending on the order of stimulus presentation, spatial configurations, and the equipment used. Notably, collinear facilitation at close distances (2λ) differed: BS was present with stereo-glasses but absent with mono-optic glasses. The research highlights the dynamic nature of BS, influenced by experimental conditions and underlying neural interactions.

This experiment acts as a more precise replication of the authors’ previous work, partially confirming prior findings. The methodology and data analysis employed are robust. Both this manuscript and the previous study (https://doi.org/10.1038/s41598-023-48380-2) would benefit from a more detailed explanation and discussion regarding the origin of inhibition at short lambda (2λ), a topic often assumed within the lateral masking literature. Clarifying the origins and mechanisms producing inhibition, based on animal physiology studies, is important. This distinction is necessary because behavioral observations (e.g., performance reduction and increased contrast threshold) and cortical inhibition mechanisms do not always align directly.

Although the authors rely on a well-established body of literature that has long supported these physiological substrates—considered foundational—it is essential to define these mechanisms precisely in relation to the experimental hypotheses and behavioral data. Providing a clear link between the physiological basis of cortical inhibition and its behavioral manifestations will enhance the scientific rigor of the discussion.

When introducing the "mixed between eyes" procedure, a clearer explanation would be helpful.

In lines 273–274, please clarify why did you use the single and not the orthogonal to calculate the TE? It is known that the single condition has more spatial uncertainty than the orthogonal. Given that you also manipulated durations, this uncertainty might vary unevenly across conditions. Specifically, spatial/temporal uncertainty can differ between the single and iso/ortho conditions. At very short durations, the presence of flankers helps localize the target more effectively than at longer durations. This effect is not due to lateral interactions but to spatial/temporal uncertainty, which varies differently across conditions. If the single condition varies across durations due to this, it could affect all TE calculations. Of course, this does not impact the comparison between monocular and binocular conditions, as the baseline would be similarly affected in both cases.

Aside from this suggestion, I have no further comments. Congratulations on a well-executed study.

Author Response

Yassin, Lev, Polat. Binocular summation. ‘Brain sciences’ review Nov 14, 2024.

Reviewer's comments

Reviewer 1

  1. Clarifying the origins and mechanisms producing inhibition, based on animal physiology studies, is important. This distinction is necessary because behavioral observations (e.g., performance reduction and increased contrast threshold) and cortical inhibition mechanisms do not always align directly. Although the authors rely on a well-established body of literature that has long supported these physiological substrates—considered foundational—it is essential to define these mechanisms precisely in relation to the experimental hypotheses and behavioral data. Providing a clear link between the physiological basis of cortical inhibition and its behavioral manifestations will enhance the scientific rigor of the discussion.

Thank you very much for your valuable and thoughtful comments and suggestions. We believe that the revised manuscript is easier to understand and will benefit the vision community. That’s why we devoted a significant part in the Discussion trying to understand and explain this issue (please see lines 900-953, P.23-24 in the "Discussion" section).

  1. When introducing the "mixed between eyes" procedure, a clearer explanation would be helpful.

Thank you for bringing this up. We are aware of this issue, but apparently, it is not clarified enough in the "Methods" section. Now we have provided a clearer explanation for the "mixed between eyes" procedure, in the "Methods" section.

By utilizing the term "mixed between eyes" procedure, we mean that across trials, the eyes (Right, Left, and Binocular) were displayed randomly, and by mixed trials (see "Binocular Testing" subsection lines 172-174, P.4 in the "Methods" section).

It means randomly across trials (intermixed). We have now clarified this issue in the "Binocular Testing" sub-section in the "Methods" section (see lines 168-177, P.4). Also, in the "Abstract" section (see lines 15-17, P.1)

  1. In lines 273–274, please clarify why did you use the single and not the orthogonal to calculate the TE? It is known that the single condition has more spatial uncertainty than the orthogonal. Given that you also manipulated durations, this uncertainty might vary unevenly across conditions. Specifically, spatial/temporal uncertainty can differ between the single and iso/ortho conditions. At very short durations, the presence of flankers helps localize the target more effectively than at longer durations. This effect is not due to lateral interactions but to spatial/temporal uncertainty, which varies differently across conditions. If the single condition varies across durations due to this, it could affect all TE calculations. Of course, this does not impact the comparison between monocular and binocular conditions, as the baseline would be similarly affected in both cases.

Thank you for bringing this up. We are aware of this issue and therefore we included a test of the orthogonal condition for comparison with the target alone (TE) and collinear 3λ, since we found that there was no significant difference in the contrast detection threshold between the single target and the orthogonal configuration at target-flanker separations of 2 & 3λ (p=0.52, p=0.31, using Tukey’s post-hoc analysis after 2-way ANOVA, respectively). It indicated that the lateral interactions at the orthogonal configuration do not affect the response to single target (because there is no effect of suppression or facilitation). Also note that there was no difference between the results of collinear and orthogonal conditions. Therefore, we decided to use the single target contrast detection threshold and not the orthogonal to calculate the threshold elevation (TE) as it used in our lab’s prior studies12,21,25–32.

Furthermore, please note that each stimulus display included four peripheral high-contrast crosses, marking the interval presentation of the target stimulus. We emphasize this in the “Methods” section (see lines 254-256, P.6), which would have obviated such uncertainty. Therefore, we believe that this approach yielded reliable data.

Reviewer 2 Report

Comments and Suggestions for Authors

see attached file

Comments on the Quality of English Language

ok

Author Response

 Yassin, Lev, Polat. Binocular summation. ‘Brain sciences’ review Nov 14, 2024.

Reviewer's comments

Reviewer 2

  1. Introduction (line 28)/Abstract (line 7). An improvement of 41% is expected on the basis of probability summation between two independent sources of information, as root (2) = 1.41. Therefore the lower figure given for the effect of binocular summation is NOT evidence for physiologically meaningful interactions, ‘awareness’, or processing of any other sort. (Thresholds for a click and a simultaneous tap would obey the same rule.) This needs to be clarified in the Abstract, and kept in mind throughout.

Thank you for bringing this up. We are aware of this issue, but apparently, it is not clarified enough in the Abstract and in the Introduction section. Now we have clarified that early studies of BS found an improvement of a factor of about 1.4 (empirically), leading to models suggesting a quadratic summation of the two monocular inputs (√2) in these sections (see lines 8-10 in the Abstract section and lines 32-33 in the Introduction section). The reviewer is correct; we added the word "may" (see line 7 in the Abstract section).

  1. Remark: The upper figure, 60%, indicates a small advantage over probability summation, but this only occurs in some cases, and so in my opinion is unlikely to reflect a consistent, higher-level effect of ‘awareness’ in the general sense (being aware of what one is seeing, as opposed to being blind to it). Perhaps a phrase such as ‘awareness of which eye is being stimulated may modulate binocular summation‘ would be less flamboyant than line 7, but more accurate. I introduced ‘may’ because there is no direct test of awareness in the paper, so this is a supposition, not a proven result. Other possibilities include that binocular neurons are sensitive to the period of stimulation of each eye. Indeed, ideas like this one appear late in the paper (lines 743 – 760; and 785). Anyway, it is widely reported that subjects cannot report which eye is stimulated when only one is, so an explanation in terms of ‘awareness’, even if true, would be highly controversial, needing very strong proof.

The reviewer is correct; we hypothesized that BS ‘may’ be affected by awareness because there is no direct test of awareness in the paper, so this is a supposition, not a proven result. In addition, when we reported that subjects cannot report which eye is stimulated when using dichoptic glasses, but they can when exposing only one eye, it is an interpretation in terms of ‘maybe it indicates awareness’; it would be highly controversial, needing very strong proof; this calls for further investigation in the future. Thank you for these comments. We clarified it throughout the paper.

  1. Page 1, line 40 “the perception of a Gabor patch” …. I think the term ‘perception’ here should rather be ‘detection’ (as opposed to the identification of a detected target, as in crowding)?

The reviewer is correct; we have now replaced the term "perception" to "detection" (see line 44, P.1).

  1. Page 2, line 51: the authors distinguish between ‘local’ (monocular within each eye) and ‘interocular’ forms of suppressive lateral interactions. I am unclear about this contrast; it crosses levels. Also, there is plenty of ‘local’ processing in cortex, some suppressive, as stated in the previous paragraph. I suggest that the author first contrast monocular with interocular, and local with global, before continuing. Since the CRF is already discussed for area V1, how about using ‘RGC’ for the retina, then explaining what ‘local’ means ‘within each eye’ (i.e., even shorter spatial distances than in V1, as RGC interactions span on average even shorter distances than V1 ones. ) ‘Local’ interocular suppression would then refer to a portion of one eye’s visual field; ‘global’ to suppression of all of it. (If indeed this is what the authros meant.)

Thanks for this point, which enabled us to elaborate about this issue. Local in physiology and psychophysics, is usually related to interactions within the vicinity of the receptive field (or perceptive field in psychophysics). Therefore, vicinity may be valid if the interactions induced from very nearby receptive fields. In this sense, lateral inhibition is largely considered as local1–6. In contrast, there are long-range interactions between neurons which may be mainly excitatory7–20. Taken these notions, local inhibition (suppression) may be found mainly at the monocular level while suppression between eyes (inter-ocular) is considered as longer-range, thus not local (between ocular dominance rather than within ocular dominance), (see lines 64-71, P.2).

  1. Note that ‘wavelength’ and ‘local’ imply different relations to distance. A low spatial frequency (giant) gabor patch and a high spatial frequency one (tiny) both obey the 2-lambda to 3-lambda distinction. However, RGCs in the retina and CRFs in cortex are defined by visual angle, as are ‘long-range’ versus ‘short-range’ (‘local’) connections. Thus in the gabor case, distance is inverse with frequency, but in the case of connections, it is fixed, independently of frequency. The logic of the research depends on lamba, not on visual angle, so the references to CRFs need to be placed in this context. There is no reason why CRFs should not be defined in terms of their preferred spatial frequency, rather than visual angle, so this is just a matter of clarification, although standard practice in physiology is to use visual angle. However, RGCs can only be defined by visual angle, so the authors’ definition of ‘local’ as ‘monocular within each eye’ cannot be defined by lambda. Some re-thinking, as well as re-writing, is needed here.

The reviewer is right in the distinction between size in physiology and size in terms of spatial frequency. We agree that the RGC and LGN receptive fields cannot be defined in terms of spatial frequency and we clarified that the meaning is to CRF in V1.  It’s true that the best way to define the size of receptive fields in physiology is in terms of the visual angle. In the early study of lateral interactions in humans21  it was demonstrated that even when using two Gabor functions and doubling their spatial frequency, and the size of the perceptive field in visual angle is doubled, the maximal facilitation is still found at collinear 3 wavelengths (λ). This relationship was confirmed in many later studies. Later physiological study22 in cat's V1 confirmed this finding; different sizes of receptive fields that were measured in visual angles show facilitation in agreement with the psychophysical studies. We have now clarified these important points in the "Introduction" section (see lines 79-86, P.2).

  1. Page 2, line 76; I cannot imagine why the previous material, which is interesting, bears on the “participant's awareness” of which eye is tested. I don’t think this is really an issue; the experimental question stands firm, anyway. But how could awareness matter to the facts listed in lines 62 to 75? Moreover, as already remarked, the subject does not know which eye is stimulated, even though the machinery of stereopsis requires that the brain ‘knows’ which eye is which, when interpreting disparities. I recall that H. Ono has demonstrated a complete lack of awareness of the eye of origin.

Thanks, indeed, our started point stems from the statement that participants have no knowledge of the eye of origin. Therefore, when using the dichoptic glasses this is the case. However, when occluding one eye, there is lack of information that is received from the occluded eye, and sometimes participants perceived rivalry between eyes. That is why we did not use a black occluder. This phenomenon may suggest that some sort of awareness may lead to this rivalry and may reduce the BS. We have now clarified this issue in the paper (see lines 107-112, P.3).  

  1. Page 2; line 92: the authors “hypothesized that since BS is not uniform, it may depend on the equipment utilized to control the monocular and binocular presentations.” This type of hypothesis is suitable for a Journal devoted to methods, but not a science-oriented Journal. The hypothesis needs to be recast as a theoretical issue, from which the particular method (e.g., relatively rapid alternation between the eyes versus very slow alternation) can be deduced. Only then does the type of equipment enter in.

Thanks, we have now reworded the text (see lines 127-129, P.3).

  1. Page 3, line 130, and later usage (e.g., line 486). I found the phrases ‘by mixed trials’ and ‘non-mixed between the eyes’ opaque. These are not standard terms. One reading is that various types of trials were either blocked or intermixed. This language would be clear to anyone in psychophysics, and can be explained briefly for others. Another reading is that different stimuli in the two eyes were mixed together – for example, 2-lambda flanks in only one eye. Often supression is demonstrated by placing a mask in one eye and the target in the other eye. But this would be dichoptic (the standard term). My suggestion is to use ‘blocked’ versus ‘intermixed’, if this is meant, and describe any differences between the stimuli to the two eyes as ‘dichoptic’, if there is any.

Thank you. Now we have used the terms ‘blocked’ versus ‘intermixed’, and we described the differences between the stimuli to the two eyes as ‘dichoptic’. We have now clarified this issue in the "Binocular Testing" subsection in the "Methods" section (see lines 168-177, P.4). Also, in the "Abstract" section (see lines 15-17, P.1)

  1. Line 131: ‘randomly and by mixed trials’. Does this mean ‘randomly across trials ‘ (i.e., ‘intermixed’ rather than ‘blocked’)? Or, is ‘random’ one condition and ‘mixed’ another condition?

It means randomly across trials (intermixed). We have now clarified this issue in the "Binocular Testing" subsection in the "Methods" section (see lines 168-177, P.4). Also, in the "Abstract" section (see lines 15-17, P.1)

  1. Page 4, line 148. I think the authors should explain in the Introduction why collinear versus orthogonal mattters, so that this doesn’t come up as a surprise variable in the Methods. Firstly, the importance of this has been studied over several decades as it relates to the type of supressive surround interaction. Therefore there is a theoretical background that can be made use of (and I am sure is vivid to the authors.) Secondly, explain how does collinear versus orthgonal relate to the main issue, namely, local versus interocular. A similar remark applies to contrast (line 197).

Hubel & Wiesel explored the classical receptive fields23,24 (CRFs) of simple cells in V1 area of the visual cortex. They found that the cells tuned selectively for location, orientation, and spatial frequency, forming the fundamental units of analysis. However, it was found that RF processing is affected by feedforward stimuli, by lateral interactions (excitatory or inhibitory signals via the horizontal connections between the ocular dominance columns found in V1) and by feedback processing. The accumulated results show that facilitation exists for collinear but not for orthogonal configuration. We have now clarified these important points in the "Introduction" section (see lines 53-60, P.2).

  1. Query: is ‘contrast’ defined by the sine-wave, or the carrier, or is it peak-trough? What is the maximum possible contrast 100% or more (depends on the definition)? Were all the gabor patches in the same phase, as in Fig. 2, or not? Please explain in the main text.

The stimulus contrast is defined by the Michelson formulation: I max- I min / I max + I min. the peak value is 127, which is considered as the maximum possible contrast (100%). All the Gabor patches were in the same phase, as in Fig. 2. We have now clarified these important points in the "The stimuli" subsection in the "Methods" section (see lines 256-258, P.6).

  1. Page 5, Line 192: sigma is missing in the pdf.

, see line 239, P.6. (λ = σ = 0.21º) Thanks, now we have clarified that

  1. Fig. 2, legend. I suggest adding ‘as illustrated in panel B ‘at the end of line 214.

Thanks, now we added the sentence "as illustrated in panel B" at the end of line 274, P.7 (see Fig. 2, legend) as the reviewer suggested.

  1. Remark: the single target condition has greater spatial uncertainty than the ‘lateral masking’ condition, which effect could reduce the difference in thresholds between them. Continuously presented crosses, shown just above and just below the single target, would have obviated any such uncertainty.

Thanks, the reviewer is correct. We dealt this issue. Each stimulus display included four peripheral high-contrast crosses, marking the interval presentation of the target stimulus, which would have obviated any such uncertainty that was mentioned above. We emphasize this in "The stimuli" subsection in the "Methods" section (see lines 254-256, P.6). Moreover, we used an additional control in which orthogonal flankers were positioned at 3λ, showing that the results of a single target and the orthogonal condition were not significantly different (p=0.31, using Tukey’s post-hoc analysis after 2-way ANOVA, respectively).

  1. Fig. 4 legend; should define ‘threshold elevation’ as log10 ( BIN or MON threshold / single target control threshold), calculated separately for each of the 4 durations. Then, state explicitly that raw threshold increased as duration decreased (Fig. 3), but did so equally for BIN and MON, as for the single target control, so the differences were roughly flat across presntation time (Fig. 4). This is obvious on reflection, and is clearly shown in Fig. S3, but should be stated in the main text to help the reader.

(Note: the Fig 4 y-axis should state log10 or log2, as the test earlier refers to log2, but log10 is standard.)

Thank you for these comments. We have now clarified these important points in the main text to help the readers (see lines 330-337, P.8), and in the capture of Fig.4 (see lines 478-480, P.12). In addition, we devoted a significant part in the "Results" section (please see lines 374-394, P.9-10) trying to explain that the raw threshold increased as the presentation time decreased (Fig. 3) as is clearly shown also in Fig. S3 in the "Supplementary Material" section.

  1. Probability summation implies that the BIN thresholds should be root(2) better than the MON thresholds. For the 2 lambda Gabors, this is true at 120 ms, since log10(root(2))=0.15, as should be clarified for the reader. Curiously, at 80 and 40 ms, the BIN thresholds fall towards the MON ones, as if some of the subjects are seeing out of one eye only. It might be possible to determine which subjects, as the authors have acuities for both eyes; briefer Gabors might be less visible to the weaker eye.

It’s true that the BIN response for short presentation times resembles the MON level. This is also demonstrated in the study25 of Lev et al, 2021. It’s indeed challenging to predict or determine from the monocular information, such as visual acuity regarding which eye contributes to the results. We ensured that the visual acuity (VA) between eyes is not different and rechecked the data now and found that there was no difference between the right and left eye regarding the VA. The participants had healthy eyes, a visual acuity of 6/6 (Log-Mar 0) or better in each eye, and there was no difference between eyes even one-line, in individuals whose eyes were fully corrected and with no ocular disease or major phoria or amblyopia (see lines 218-221 in the "Participants" subsection in the "Methods" section and Table 1 in the "Supplementary Material" section for more details). In addition, we found that there was no significant difference in the contrast detection threshold between the eyes for each condition during each presentation time during the different experiments (see lines 324-329 in the "Procedure" subsection in the "Methods" section), indicating that the performance of both eyes was similar; therefore, "monocular" in this study refers to the mean monocular (see Table 2 in the "Supplementary Material" section for statistical information).

 We now elaborated on this issue in the Discussion (please see lines 875-899, P.22 in the "Discussion" section).

  1. Page 13, line 483. I suspect that ‘depending on’ rather than ‘relying on’ is meant.

The reviewer is correct; we have now replaced the term "relying on" to "depending on" (see line 564, P.15).

  1. Fig. 7. I do not understand the relation between 7A and 7B. In 7A, it appears that MON and BIN thresholds differ for the 2-lamba stimuli by 0.2 log units, in the direction of binocular suppression. However, in 7B, the COLL-2 data show no binocular suppression (or summation). I am unclear about the factors responsible for this difference. It is not the sequence of durations, as this was the same (from long to short) in both experiments. So, was it the use of lateral masking, the glasses, the practice, or the change from intermixed to blocked trials? The authors discuss trial order, but data in Fig. 7A are a near replication of those in Fig. 4, for both 2- and 3-lambda, and the only difference I can find is that in fig. 4, trials were intermixed at random, and in fig. 7A, they were blocked (if I understand the terminology). The conclusion follows that the order of stimulation across trials, random or blocked, does not matter. However, the authors flatly contradict this conclusion, when referring to fig. 7B, so I am perplexed.

Thank you for these comments, the reviewer is right. In Fig. 4, the order of stimulation across trials was intermixed at random for the eye presentation; however in Fig. 7, the order of stimulation across trials was blocked for the eye presentation. We have now clarified this issue in the captures of Fig.4&7, respectively and in the main text in the "Results" section in the relevant statement.

Figure 7A presents binocular collinear suppression for the collinear 2λ condition at all the presentation times; consistently Figure 7B presents the absence of a BS effect for the collinear 2λ condition at all presentation times. However, Figure 4 indicates that under the collinear 2λ condition, there was no effect of binocular suppression at all presentation times (see Figure 4). We have now clarified this issue in the main text in the "Results" section in the relevant statement.

The results for the collinear 3λ condition were uniform in Figures 4 & 7, which indicated that clear collinear facilitation was observed for both monocular and binocular presentations at all the presentation times.

We have now clarified this issue in the paper.

  1. Fig. 6 versus fig.8. These came out as different in the pdf, so I copied both figures to powerpoint and stretched the y-axes to be equal to each other. I then copied one figure onto the other. As far as I can see, the 3-lambda data overlap for every subject except the one at the very top of the lollipop in fig. 8. I thnik the authors should consider the hypothesis that this one subject explains the 3-lambda difference, rather than there being a systematic shift in the distribution. The number of subjects is small enough for such an effect (which changes the skew) to violate the nul hypothesis, especially as different subjects were run in the two experiments. I think this is a more plausible hypothesis than the one offered, because probablity summation does such a good job of explaining the 3-lambda data.

Thanks for this important point, which enabled us to elaborate. Importantly, although initially it seems that the collinear 3λ data overlap for every subject except the one at the very top of the lollipop in Fig. 8 which may suggest that this one subject explains the difference for the collinear 3λ condition, rather than there being a systematic shift in the distribution. Now we double-checked this in the BS ratio data for the collinear 3λ condition for Fig.8 and we found that the mean was 1.72 and the median was 1.76 (including the participant with the high ratio of BS = 2.55). When we calculated the mean and the median without including this subject we found that the mean was 1.67 and the median was 1.73. In addition, we performed a welch two sample t-test with equal variance (Paired Student’s t-test) to evaluate the statistical differences between the two data sets for the collinear 3λ condition (including this subject and without including this subject). We found that there was no significant difference between the 2 data sets (p=0.68, paired, two-tailed, t-test); So we decide to keep and including this subject in the data and clarified it in the relevant statement in the paper. Now we have clarified this issue in the text under Fig.8 (see line 649-665, P.17). Furthermore, the normality of residuals and the homogeneity of variance assumptions were assessed graphically utilizing diagnostic plots, in our study all data points were approved as not being outliers (see lines 352-354, P.9 in the "Data and statistical analysis" sub-section in the "Methods" section).

  1. Spelling things out would somewhat increase the length of the paper. However, this can be compensated for by placing the text describing the statistics in the supplementary material, as the effects are large (relative to the standard errors, as nicely shown in the figures) and are clear to see. (Indeed, I suggest replacing the statistical text not only with more explication, but also with more of the raw data in the supplementary materials, as these are far more informative. However, I acknowledge that Journals have started to insist on more statistics, even for psychophysical papers where they are mostly performative, so this advice may violate Editorial policy.)

Thank you for these comments; we agree with the reviewer that the need for detailed statistics may dilute the highlight of the main results, especially when they are easily noticed in the figures. However, since many journals require detailed statistics in the description of the results, we decided to keep the text describing the statistics as it has been elaborated in the paper.

For more raw data: We confirmed at the "Data Availability Statement" section that "The datasets used and/or analyzed during the current study are available from the corresponding author on reasonable request".

References

  1. Liu, X. et al. From Receptive to Perceptive Fields: Size-Dependent Asymmetries in Both Negative Afterimages and Subcortical On and Off Post-Stimulus Responses. J. Neurosci. 41, 7813–7830 (2021).
  2. Harvey, B. M. & Dumoulin, S. O. The Relationship between Cortical Magnification Factor and Population Receptive Field Size in Human Visual Cortex: Constancies in Cortical Architecture. J. Neurosci. 31, 13604–13612 (2011).
  3. Spillmann, L. The Hermann Grid Illusion: A Tool for Studying Human Perceptive Field Organization. Perception 23, 691–708 (1994).
  4. Yazdanbakhsh, A. & Gori, S. A new psychophysical estimation of the receptive field size. Neurosci. Lett. 438, 246–251 (2008).
  5. Jones, J. P. & Palmer, L. A. An evaluation of the two-dimensional Gabor filter model of simple receptive fields in cat striate cortex. J. Neurophysiol. 58, 1233–1258 (1987).
  6. Spillmann, L., Dresp-Langley, B. & Tseng, C. Beyond the classical receptive field: The effect of contextual stimuli. J. Vis. 15, 7 (2015).
  7. Adini, Y., Sagi, D. & Tsodyks, M. Excitatory-inhibitory network in the visual cortex: Psychophysical evidence. Proc. Natl. Acad. Sci. U. S. A. 94, 10426–10431 (1997).
  8. Stettler, D. D., Das, A., Bennett, J. & Gilbert, C. D. Lateral Connectivity and Contextual Interactions in Macaque Primary Visual Cortex. Neuron 36, 739–750 (2002).
  9. Polat, U. Functional architecture of long-range perceptual interactions. Spat. Vis. 12, 143–162 (1999).
  10. Mizobe, K., Polat, U., Pettet, M. & Kasamatsy, T. Facilitation and suppression of single striate-cell activity by spatially discrete pattern stimuli presented beyond the receptive field. Vis. Neurosci. 18, 377–391 (2001).
  11. Chen, C.-C. & Tyler, C. W. Excitatory and inhibitory interaction fields of flankers revealed by contrast-masking functions. J. Vis. 8, 10 (2008).
  12. Polat, U. & Sagi, D. Temporal asymmetry of collinear lateral interactions. Vision Res. 46, 953–960 (2006).
  13. Grinvald, A., Lieke, E., Frostig, R. & Hildesheim, R. Cortical point-spread function and long-range lateral interactions revealed by real-time optical imaging of macaque monkey primary visual cortex. J. Neurosci. 14, 2545–2568 (1994).
  14. Gerard-Mercier, F., Carelli, P. V., Pananceau, M., Troncoso, X. G. & Frégnac, Y. Synaptic correlates of low-level perception in V1. J. Neurosci. 36, 3925–3942 (2016).
  15. Kapadia, M. K., Westheimer, G. & Gilbert, C. D. Spatial Distribution of Contextual Interactions in Primary Visual Cortex and in Visual Perception. J. Neurophysiol. 84, 2048–2062 (2000).
  16. Nelson, J. I. & Frost, B. J. Experimental Brain Research Intracortical Facilitation among Co-Oriented, Co-Axially Aligned Simple Cells in Cat Striate Cortex. Exp Brain Res vol. 61 (1985).
  17. Kapadia, M. K., Ito, M., Gilbert, C. D. & Westheimer, G. Improvement in Visual Sensitivity by Changes in Local Context: Parallel Studies in Human Observers and in V1 of Alert Monkeys. Neuron vol. 15 (1995).
  18. Ts’o, D., Gilbert, C. & Wiesel, T. Relationships between horizontal interactions and functional architecture in cat striate cortex as revealed by cross-correlation analysis. J. Neurosci. 6, 1160–1170 (1986).
  19. Sincich, L. C. & Blasdel, G. G. Oriented axon projections in primary visual cortex of the monkey. J. Neurosci. 21, 4416–4426 (2001).
  20. Gilbert, C. D. & Wiesel, T. N. Columnar specificity of intrinsic horizontal and corticocortical connections in cat visual cortex. J. Neurosci. 9, 2432–2422 (1989).
  21. Polat, U. & Sagi, D. Lateral interactions between spatial channels: Suppression and facilitation revealed by lateral masking experiments. Vision Res. 33, 993–999 (1993).
  22. Polat, U., Mizobe, K., Pettet, M. W., Kasamatsu, T. & Norcia, A. M. Collinear stimuli regulate visual responses depending on cell’s contrast threshold. Nat. 1998 3916667 391, 580–584 (1998).
  23. Hubel, D. H. & Wiesel, T. N. Receptive fields and functional architecture of monkey striate cortex. J. Physiol. 195, 215–243 (1968).
  24. Hubel, D. H. & Wiesel, T. N. Receptive fields, binocular interaction and functional architecture in the cat’s visual cortex. J. Physiol. 160, 106 (1962).
  25. Lev, M., Ding, J., Polat, U. & Levi, D. M. Nearby contours abolish the binocular advantage. Sci. Reports 2021 111 11, 1–17 (2021).
  26. Yassin, M., Lev, M. & Polat, U. Space, time, and dynamics of binocular interactions. Sci. Rep. 13, 21449 (2023).
  27. Yassin, M., Lev, M. & Polat, U. Dynamics of the perceptive field size in human adults. Vision Res. 224, 108488 (2024).
  28. Benhaim-Sitbon, L., Lev, M. & Polat, U. Binocular fusion disorders impair basic visual processing. Sci. Rep. 12, 12564 (2022).
  29. Benhaim-Sitbon, L., Lev, M. & Polat, U. Extended perceptive field revealed in humans with binocular fusion disorders. Sci. Rep. 13, 6584 (2023).
  30. Serero, G., Lev, M. & Polat, U. Distorted optical input affects human perception. Sci. Rep. 10, 11527 (2020).
  31. Polat, U. & Sagi, D. The architecture of perceptual spatial interactions. Vision Res. 34, 73–78 (1994).
  32. Lev, M. & Polat, U. Space and time in masking and crowding. J. Vis. 15, 10–10 (2015).